# Definition of the immune evasion-replication interface of rabies virus P protein

Jingyu Zhan[1,2], Angela R. Harrison[3], Stephanie Portelli[1,2], Thanh Binh Nguyen[1,2], Isshu Kojima[4], Siqiong Zheng[1,2], Fei Yan[1,2], Tatsunori Masatani[5], Stephen M. Rawlinson[3], Ashish Sethi[1,2], Naoto Ito[5], David B. Ascher[1,2], Gregory W. Moseley[3]*, Paul R. Gooley[1,2]*

1 Department of Biochemistry and Pharmacology, University of Melbourne, Parkville, Australia, 2 Bio21 Molecular Science and Biotechnology Institute, University of Melbourne, Parkville, Australia, 3 Department of Microbiology, Biomedicine Discovery Institute, Monash University, Clayton, Australia, 4 Joint Graduate School of Veterinary Medicine, Kagoshima University, Kagoshima, Japan, 5 Laboratory of Zoonotic Diseases, Joint Department of Veterinary Medicine, Faculty of Applied Biological Sciences, Gifu University, Gifu, Japan

* prg@unimelb.edu.au (PRG); greg.moseley@monash.edu (GWM)

**Data Availability Statement:** All relevant data are within the manuscript and its Supporting Information files.

**Funding:** This research was supported by National Health & Medical Research Council Australia

## Abstract

Rabies virus phosphoprotein (P protein) is a multifunctional protein that plays key roles in replication as the polymerase cofactor that binds to the complex of viral genomic RNA and the nucleoprotein (N protein), and in evading the innate immune response by binding to STAT transcription factors. These interactions are mediated by the C-terminal domain of P ($P_{CTD}$). The colocation of these binding sites in the small globular $P_{CTD}$ raises the question of how these interactions underlying replication and immune evasion, central to viral infection, are coordinated and, potentially, coregulated. While direct data on the binding interface of the $P_{CTD}$ for STAT1 is available, the lack of direct structural data on the sites that bind N protein limits our understanding of this interaction hub. The $P_{CTD}$ was proposed to bind *via* two sites to a flexible loop of N protein ($N_{pep}$) that is not visible in crystal structures, but no direct analysis of this interaction has been reported. Here we use Nuclear Magnetic Resonance, and molecular modelling to show N protein residues, Leu381, Asp383, Asp384 and phosphor-Ser389, are likely to bind to a 'positive patch' of the $P_{CTD}$ formed by Lys211, Lys214 and Arg260. Furthermore, in contrast to previous predictions we identify a single site of interaction on the $P_{CTD}$ by this $N_{pep}$. Intriguingly, this site is proximal to the defined STAT1 binding site that includes Ile201 to Phe209. However, cell-based assays indicate that STAT1 and N protein do not compete for P protein. Thus, it appears that interactions critical to replication and immune evasion can occur simultaneously with the same molecules of P protein so that the binding of P protein to activated STAT1 can potentially occur without interrupting interactions involved in replication. These data suggest that replication complexes might be directly involved in STAT1 antagonism.

grants (https://www.nhmrc.gov.au/) 1125704 (G. W.M and P.R.G), 1079211, 1003244, 1160838 (G. W.M), 1174405 (D.B.A and T.B.N), Australian Research Council discovery project grant (https://www.arc.gov.au/) DP150102569 (G.W.M), Meigunyah Trust Grimwade Fellowship (G.W.M), Australian Government Research Training Program Scholarship (A.R.H) and Melbourne Research Scholarships (J.Z. and S.P.). The funders had no role in study design, data collection and analysis, decision to publish, or preparation of the manuscript.

**Competing interests:** The authors have declared that no competing interests exist.

## Author summary

For viruses to infect cells and generate progeny, they must be able to mediate replication, while simultaneously evading the innate immune system. Viruses with small genomes often achieve this through multifunctional proteins that have roles in both replication and immune evasion, such as the phosphoprotein (P protein) of rabies virus. P protein is an essential cofactor in genome replication and transcription, dependent on the well-folded C-terminal domain ($P_{CTD}$), which binds to the nucleoprotein (N protein) when complexed with RNA. The $P_{CTD}$ can also bind and antagonize signal transducers and activators of transcription (STAT) proteins, that are essential for activating antiviral mechanisms. Here we show using Nuclear Magnetic Resonance spectroscopy and cell-based assays, that the STAT1-binding and N-binding interfaces are proximal but, nevertheless, it appears that the same molecule of $P_{CTD}$ can simultaneously bind STAT1 and N protein. These data suggest that P-protein-STAT1 interaction, critical to immune evasion, can occur without interrupting interactions underlying replication, and so replication complexes might be directly involved in STAT1 antagonism.

## Introduction

Rabies virus (RABV) is a member of the *Lyssavirus* genus (family *Rhabdoviridae*, order *Mononegavirales*) [1], which causes acute neurological disease in humans with a c. 100% case fatality rate, resulting in > 59,000 deaths/year world-wide [2–4]. In common with other viruses in the order *Mononegavirales*, RABV has a non-segmented single-stranded negative sense RNA genome [5,6], which is encapsidated by the nucleoprotein (N protein) to form the helical N-RNA ribonucleoprotein complex. This complex serves as the template for viral genome transcription and replication by the RNA-dependent RNA polymerase complex [7], which is composed of the enzymatic L protein and non-catalytic polymerase cofactor phosphoprotein (P protein) [8,9]. In addition to acting as the RNA-dependent RNA polymerase, L protein also catalyses mRNA capping and polyadenylation. Importantly, L is a processive enzyme that must remain attached and proceed along the N-RNA template [9,10]. However, L protein does not bind to the N-RNA complex to access genomic RNA directly, and so is absolutely dependent on P protein which mediates attachment to the N-RNA template [11]. These basic transcription and replication mechanisms are shared by all members of the *Mononegavirales* order. The viral polymerase complex P/L together with N-RNA accumulates in RABV-infected cells, forming cytoplasmic inclusion bodies termed Negri bodies (NBs) through liquid-liquid phase separation. These N/P rich compartments comprise viral factories where genome transcription and replication take place [12,13].

P protein is a multi-domain protein that forms key interactions with viral proteins critical to replication [14,15] and also with host factors that underlie diverse roles in the virus-host interface, particularly in immune evasion [16–21]. Cellular interactors include critical components of interferon (IFN) signalling pathways, and several elements of the host cellular trafficking machinery [22,23], which enable P protein to travel between the host cell cytoplasm and nucleus, a process implicated in immune evasion [18,24,25]. The well-structured C-terminal globular domain ($P_{CTD}$), spanning residues 186–297 of P protein [26,27], represents a critical interface for viral replication as the site of interaction with N-RNA [14,28], as well as for interactions with host-cell proteins, including several signal transducers and activators of transcription (STAT) proteins, important to immune evasion [16–18,20,21,24,29].

STATs are a family of transcription factors that are critical mediators of cytokine signalling. Among these, STAT1 is activated by phosphorylation (pY-STAT1) on a conserved tyrosine (Y701) in response to antiviral type-1 interferons (IFN-α/β), resulting in nuclear accumulation and activation of IFN-stimulated genes that can activate an antiviral state [30]. $P_{CTD}$ binds to pY-STAT1 [29] involving residues I201 to F209 and D235 to I237 [16] (S1 Fig), resulting in inhibition of pY-STAT1-DNA binding and nuclear exclusion of the P/STAT1 complex *via* a nuclear export sequence (NES) in the P protein N-terminal region [22]. The role of the $P_{CTD}$ in mediating multiple interactions including STAT1 and N protein is intriguing, as it provides a potential regulatory hub in replication and immune evasion. Thus, the nature of the interactions and how the small $P_{CTD}$ coordinates N and STAT1 is of significant interest but remains poorly understood.

Initial insights into these interactions included mutagenic studies of P protein which indicated roles of a positive patch on the $P_{CTD}$ ($_{211}$KKYK$_{214}$) in interaction with N protein [31,32], and of a hydrophobic pocket (the 'W-hole', which includes C261, W265, M287, see S1 Fig) on $P_{CTD}$ in STAT1 binding [29]. Positioning of these regions on opposite faces of $P_{CTD}$ provides a potential mechanism to bind both partner molecules simultaneously. However, the recent structural analysis of STAT1 interaction indicated that the mutations of the W-hole act indirectly, and the STAT1 interface [16] is, in fact, proximal to the proposed N protein-binding site (S1 Fig), indicative of a more complex interface.

Available crystal structures of the rabies N protein in complex with RNA (N-RNA) [33,34] show a ring of ten or eleven bi-lobal N-protein subunits, consisting of N-terminal ($N_{NTD}$) and C-terminal domains ($N_{CTD}$), with RNA bound in a groove between the $N_{NTD}$ and $N_{CTD}$. Tomography and cryo-electron microscopy of the virion of RABV [35] and the closely related vesicular stomatitis virus [36] visualized the N-RNA complex and M protein, showing that the N-RNA forms a series of inner helical turns with an outer array of M-protein. The ring and virion structures differ mostly in how the N-protein subunits relate to each other, where the subunits are closer to each other and oriented differently in the former crystal structures. These structures of RABV N protein do not include the P protein. Therefore, there is no direct structural data on the interaction of the P and N proteins. The only structural insights come from a modelling study of the N-RNA/$P_{CTD}$ complex, based on a random mutagenic screen of the $P_{CTD}$ of the lyssavirus Mokola virus (MOKV, a distantly related member of the genus) used to identify residues involved in binding to N-RNA [32], the crystal structures of RABV $P_{CTD}$ [26] and N-RNA [33,34], and small angle X-ray scattering (SAXS) data [14]. A disordered region in N (residues 376–397) is absent in the crystal structure. Removal of this loop and a region C-terminal to this loop (N-pep, residues 377–450) by trypsin digestion results in loss of binding to P protein [37–39], indicating that this region likely harbours interaction sites for the $P_{CTD}$. The $P_{CTD}$ was proposed to lie on the C-terminal top of the N-RNA ring, where disordered $N_{CTD}$ loops from adjacent N monomers act as pincers that contact the $P_{CTD}$ on two separate interfaces, the positive-charged patch and the W-hole. However, these predictions are controversial as there are no direct data supporting a role for the W-hole in N binding [31]. Importantly, as the residues of the W-hole are not conserved, the potential interaction surface would differ for P proteins of different strains, casting doubt that the W-hole forms a common interaction site [27]. Furthermore, while yeast two-hybrid mutagenic analysis performed with MOKV proteins [31] and mammalian cell-based assays performed with RABV proteins [29] were consistent with a role for the positive patch, the latter study suggested that mutations of the W-hole residues did not impair N-binding or viral replication. Although the yeast two-hybrid analysis of MOKV P identified proteins lacking N interaction with mutations in the W-hole, these also contained mutations in the positive patch. Thus, there is no direct evidence for a role of the W-hole, and the role of the positive patch is based

solely on data from mutagenesis for which potential off-target effects in $P_{CTD}$ have been clearly demonstrated [16].

Given the proximity of the positive patch to the newly identified STAT1-binding site of the $P_{CTD}$ (S1 Fig), the precise role of these sites remains unclear [16]. The role of the $N_{CTD}$ loop as the putative $P_{CTD}$ binding site is also unclear as it was not resolved in the N-RNA crystal structure, and mutagenic analysis of MOKV N did not identify any critical residues for P protein binding [31]. Specifically, mutating individually or in combination the two highly conversed acidic regions (residues $_{373}EE_{374}$ and $_{390}DEED_{393}$ of MOKV N) did not affect the interaction of N and P proteins [31]. Thus, the molecular basis of N-P protein interaction, and the relationship of STAT1/N protein interfaces underpinning replication and immune evasion remain poorly understood.

In the present study, we aimed to characterize the molecular interactions of the $P_{CTD}$-N protein interface in both partners. Using NMR titrations, we show directly that N-pep binds at the positive patch of $P_{CTD}$, and further show that it does not bind at the W-hole. Using two-dimensional line-shape NMR analysis and isothermal titration calorimetry (ITC), we measured the binding thermodynamics and kinetics of mutated proteins to identify key N residues involved in binding. Importantly, these data directly indicated that the N-binding and STAT1 binding sites [16] are proximal. However, cell-based assays indicated no competition in binding, with the $P_{CTD}$ able to interact simultaneously with N and STAT1. Thus, binding of STAT1 to P protein for immune evasion appears not to disrupt interactions with N protein involved in genome replication, suggesting that replication complexes may contribute to STAT1 antagonism.

## Results

### The C-terminal loop region of N-protein is intrinsically disordered

The N-pep as a linear peptide in solution showed resonances that were dispersed over a narrow range in the $^1H^N$-dimension (Fig 1A), indicating that N-pep is predominantly disordered, consistent with its flexibility in the crystal structure [26]. Backbone $^1H$, $^{15}N$ and $^{13}C$ resonances of N-pep (residues 363–414) were assigned using standard three-dimensional heteronuclear NMR experiments. The $^{13}C\alpha$ and $^{13}C\beta$ chemical shifts depend on local backbone geometry and are exquisitely sensitive to the presence of secondary structure, which provides a means to identify regions of transient regular secondary structure [40]. Secondary Structure Propensity (SSP) scores (Fig 1B) show that there are weak positive trends from 361 to 377 and from 402 to 410 consistent with the presence of transient helical structures (362 to 377, 402 to 411) within the N protein. The region from 378 to 401 shows a mix of weak negative and positive trends. Nevertheless, the isolated N-pep is predominantly disordered.

### Atomic level structural insights into the N-pep/$P_{CTD}$ interaction

Data on the interaction of N-RNA/P is available only from random mutagenesis [32] and a subsequent modelling study using these data [14], and from a targeted mutagenic analysis of the MOKV $P_{CTD}$ positive patch [31]. To directly assess binding sites in the $P_{CTD}$ as well as in N protein, and to gain insight into the binding kinetics, NMR chemical shift titration experiments were performed. Addition of an increasing amount of unlabelled N-pep into $^{15}N$-labelled $P_{CTD}$ resulted in a discrete perturbation (Fig 2). Most of these perturbed peaks shifted smoothly from the free state to the bound state (Figs 2B and S2), suggesting that they are experiencing fast exchange on the NMR time scale. Some of the signals which moved the most (K212 to F215) broadened significantly with the addition of N-pep and then sharpened up again sufficiently near the saturation point to be detected. The largest chemical shift

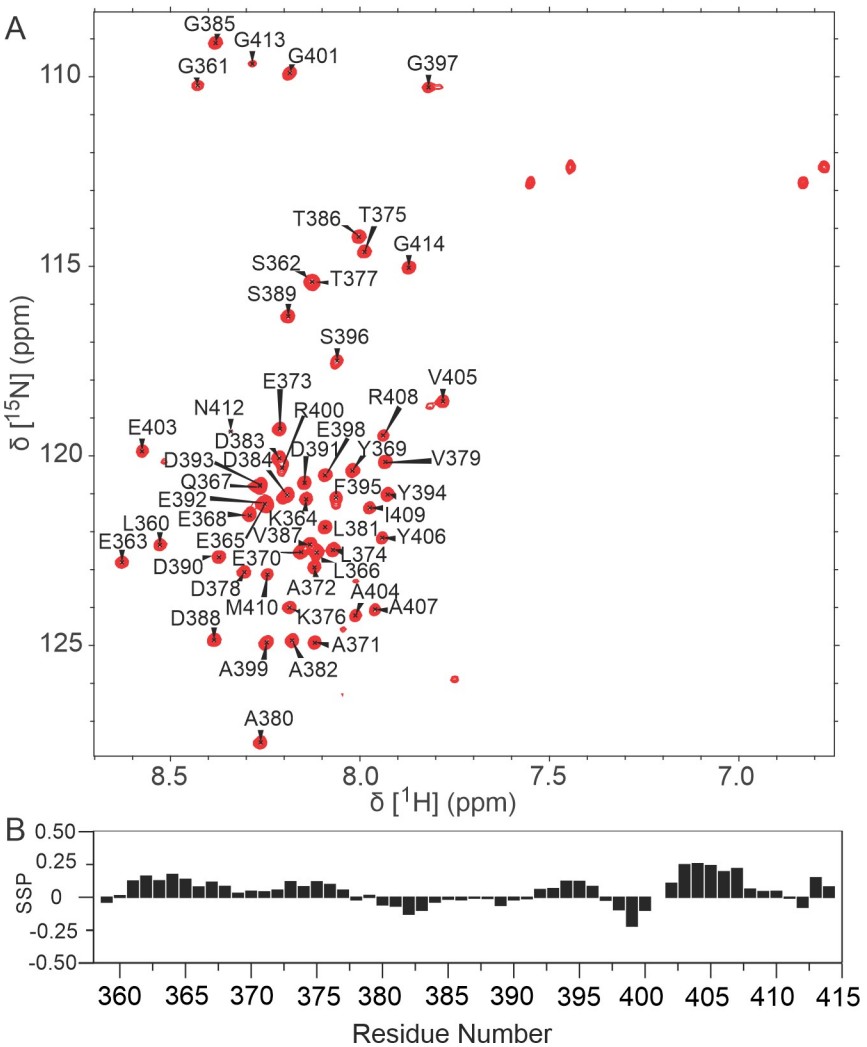

**Fig 1. N-pep backbone assignment and secondary structure propensity analysis.** (A) 700 MHz ¹H-¹⁵N HSQC spectra of ¹⁵N-labelled N-pep at pH 6.8 and 25°C. Assignment of N-pep backbone amide resonances are indicted by residue and number. Resonances from Q367 and N412 side chain on the upper right spectra are not assigned. (B) Secondary structure propensity analysis of N-pep. Secondary structure was analysed by SSP using ¹³Cα and ¹³Cβ chemical shift data. No patterns could be discerned from the N-pep chemical shifts, indicating the N-pep was predominantly unstructured.

perturbations in $P_{CTD}$ were observed within or close to a positive patch, K211, K212, K214 and R260 (Fig 2A and 2D), and the most significant resonance broadening was also observed in this region. The perturbations in the positive patch provide direct evidence that this represents the major molecular interaction site, agreeing with results of the random and site-directed mutagenesis studies that suggested roles for this region as affecting RABV and MOKV $P_{CTD}$/N binding [31,32]. Notably, several hydrophobic residues, F223 and L224 which are near the positive patch, showed significant shifts, but in contrast, the W-hole (C261, W265 and M287) on the opposite face of $P_{CTD}$ which was predicted to comprise a second N-binding site [14], based on results of random mutagenesis and the $P_{CTD}$ structure [26,32], showed smaller perturbations (CSP < 0.16 ppm, which lie within one standard deviation of the CSP of the remaining peaks, calculated at the last titration point of 1 $P_{CTD}$: 9 N-pep according to Eq 1). Importantly, the $N^{\epsilon1}/H^{\epsilon1}$ of the indole ring of W265 was not perturbed throughout the titration (Fig 2B).

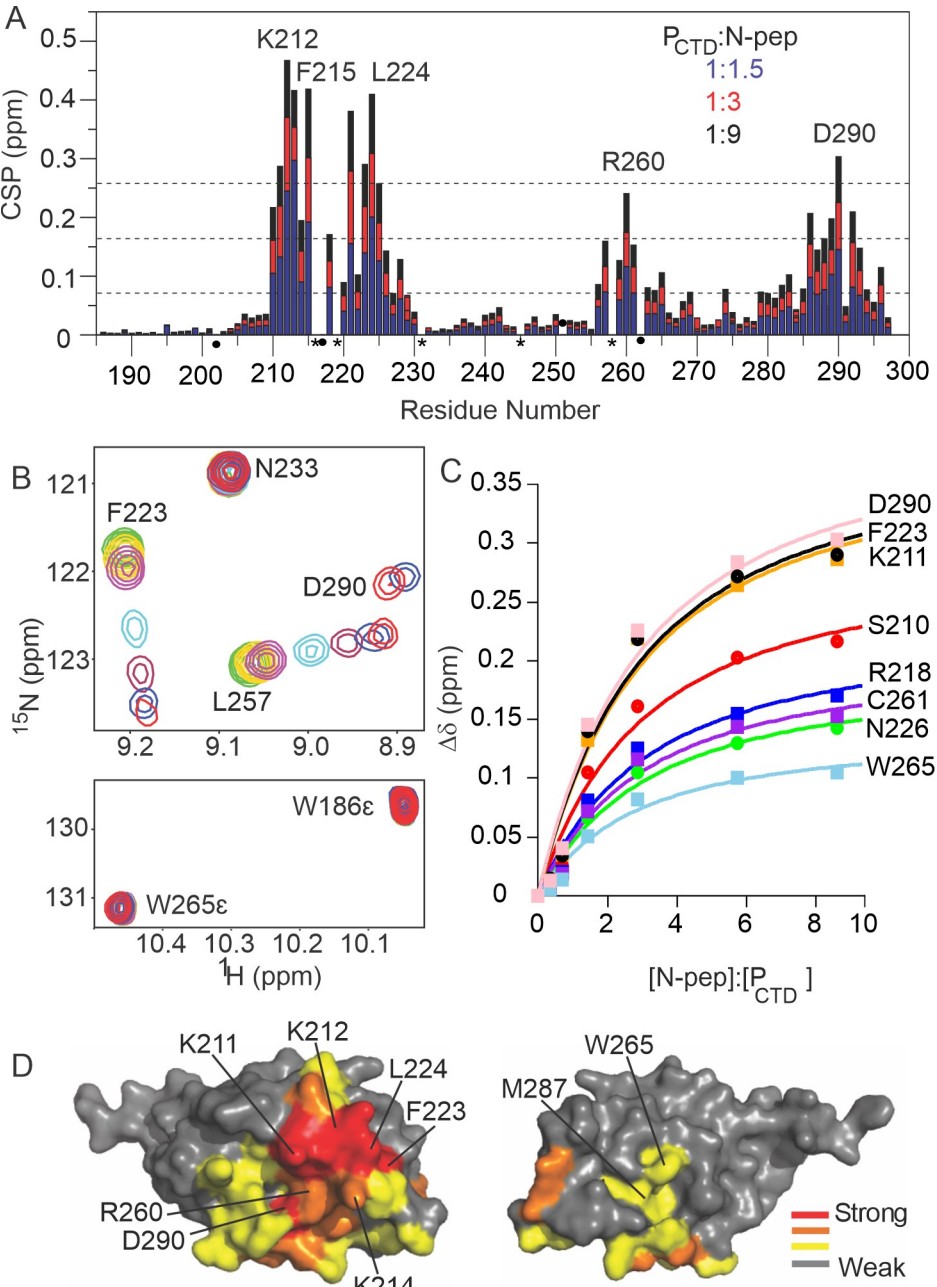

**Fig 2. Titration of $^{15}$N-labelled P$_{CTD}$ with N-pep.** (A) Plot of P$_{CTD}$ amide chemical shift changes. The blue, red and black colour corresponds to the chemical shift changes at P$_{CTD}$: N-pep molar ratios of 1:1.5, 1:3, 1:9 respectively. Dashed lines indicate the mean, first standard deviation, and second standard deviation at the last titration point. (B) Sections from $^{15}$N-P$_{CTD}$, $^{1}$H-$^{15}$N HSQC titration experiment. 7 spectra were recorded at P$_{CTD}$: N-pep molar ratios of 1:0, 1:0.4, 1:0.8, 1:1.5, 1:3, 1:6 and 1:9. Experiments were conducted at pH 6.8 and 25˚C. Green colour indicated apo form of 90 μM of $^{15}$N-P$_{CTD}$. Adding N-pep resulted in the chemical shift perturbation and reduction of intensity of F223 and L257, while N233 was not perturbed. Also, W265 indole sidechain in the W-hole was not perturbed by the addition of N-pep throughout the titration. (C) Saturation binding curves for the well-resolved $^{15}$NH resonances of the residues S210, K211, R218, F223, N226, C261, W265 and D290 which all show fits to a single binding event (K$_D$ = 246 ± 10 μM). (D) The round face (left) and the flat face (right) of P$_{CTD}$. The residues are coloured according to the chemical shift differences of the last titration point (red, two standard deviation; orange, one standard deviation; yellow, mean). Mapping the chemical shift changes onto the crystal structure of P$_{CTD}$ indicated the positive patch on the round face of P$_{CTD}$ forms the key binding site for N-pep.

Consequently, the binding event detected by the peptide NH of the W-hole residues is that occurring with the positive patch and may reflect small conformational changes to the $P_{CTD}$. All observed shifts were in a linear manner, indicative of a single binding event with a single dissociation constant. Fitting the chemical shift differences of resonances that were resolved and observable throughout the titration to a single binding site model (Figs 2C and S2), shows the affinity ($K_D$) of N-pep for $P_{CTD}$ is 246 ± 10 μM.

To map the interaction site of $P_{CTD}$ on N-pep, a complementary titration monitoring chemical shift changes in [15]N-labelled N-pep upon addition of unlabelled $P_{CTD}$ was performed. Upon the addition of $P_{CTD}$, about 20 resonances within the flexible loop region (residue 375–395) were distinctly shifted and/or broadened while the remaining 30 resonances were not affected (Fig 3A and 3B). Within the flexible loop region, eight resonances showed small but significant shifts typical of the fast exchange regime (V379, S389-F395). In the presence of less than 1-fold molar excess of $P_{CTD}$, resonances from D378, A380, A382 and D383 were broadened out and remained unobservable throughout the remainder of the titration, whereas another nine residues, T375-T377, L381 and D384-D388, broadened but persisted throughout the titration. The signal intensity within the binding interface on N-pep decreased much faster compared to that of $P_{CTD}$, suggesting that N-pep may be undergoing a conformational transition upon binding to $P_{CTD}$, and that the possible folding event occurs on the micro-millisecond time scale, based on the line broadening observed in the binding site of N-pep. The binding affinity obtained from the chemical shift perturbation is 224 ± 9 μM (Fig 3C), consistent with the titration of [15]N-labelled $P_{CTD}$ with N-pep. These data provide direct information on the residues of N protein mediating interactions with $P_{CTD}$.

## Effect of cyclization of N-pep

The micromolar affinity observed in our N-pep/$P_{CTD}$ titration experiments is weaker compared to the reported nanomolar affinity of N-RNA/$P_{CTD}$ measured by surface plasmon resonance (SPR) [14], which could be partially due to lack of constraint at the termini of the flexible loop region in the N-pep construct. Models of the N-pep binding site for $P_{CTD}$ [14] suggest that it appears as a loop in the full-length N protein, with residues T375 close to S396, which proposes the idea of cyclizing by building in a disulfide bond. To restrict and circularise N-pep we inserted two cysteines outside of the region that showed significant chemical shift perturbation on binding to $P_{CTD}$: one between E370 and A371; and the other between G401 and P402. A [1]H-[15]N HSQC titration of [15]N-labelled cyclized N-pep with unlabelled $P_{CTD}$ was performed as above, showing the perturbed peaks are experiencing exchange mostly on an intermediate time-scale regime, but fitting to a single binding site as indicated by the linearity of the peak shifts during the titration. The dissociation constant determined from the saturation binding curves is 50.6 ± 5.5 μM (Table 1), which is 4.5-fold tighter than linear N-pep (224 ± 9 μM).

We also performed two-dimensional line-shape analysis [41] of the wild type [15]N-labelled N-pep and $P_{CTD}$ NMR titration data for residues G385, D388, D390, S396 and G397 of N-pep which fitted well to a two-state binding model (S3 Fig), and yielded a dissociation constant of 215 ± 6 μM and an off-rate of 3164 ± 168 s[-1]. Line-shape analysis was also performed on [15]N-labelled cyclized N-pep and $P_{CTD}$ titration data (Table 1). Compared with the linear wild type N-pep, cyclization improved the binding affinity 4.5-fold to 47.8 ± 2.4 μM by extending the complex lifetime ($k_{off}$ = 724 ± 53 s[-1]), while the association rate ($k_{on}$) remained the same. The dissociation constants calculated from the two-dimensional line-shape analysis for both the linear and cyclized N-pep are in good agreement with the saturation binding curve analysis implying that conformational restriction of the $N_{CTD}$ loop in the full-length N protein improves the affinity of its interaction with $P_{CTD}$.

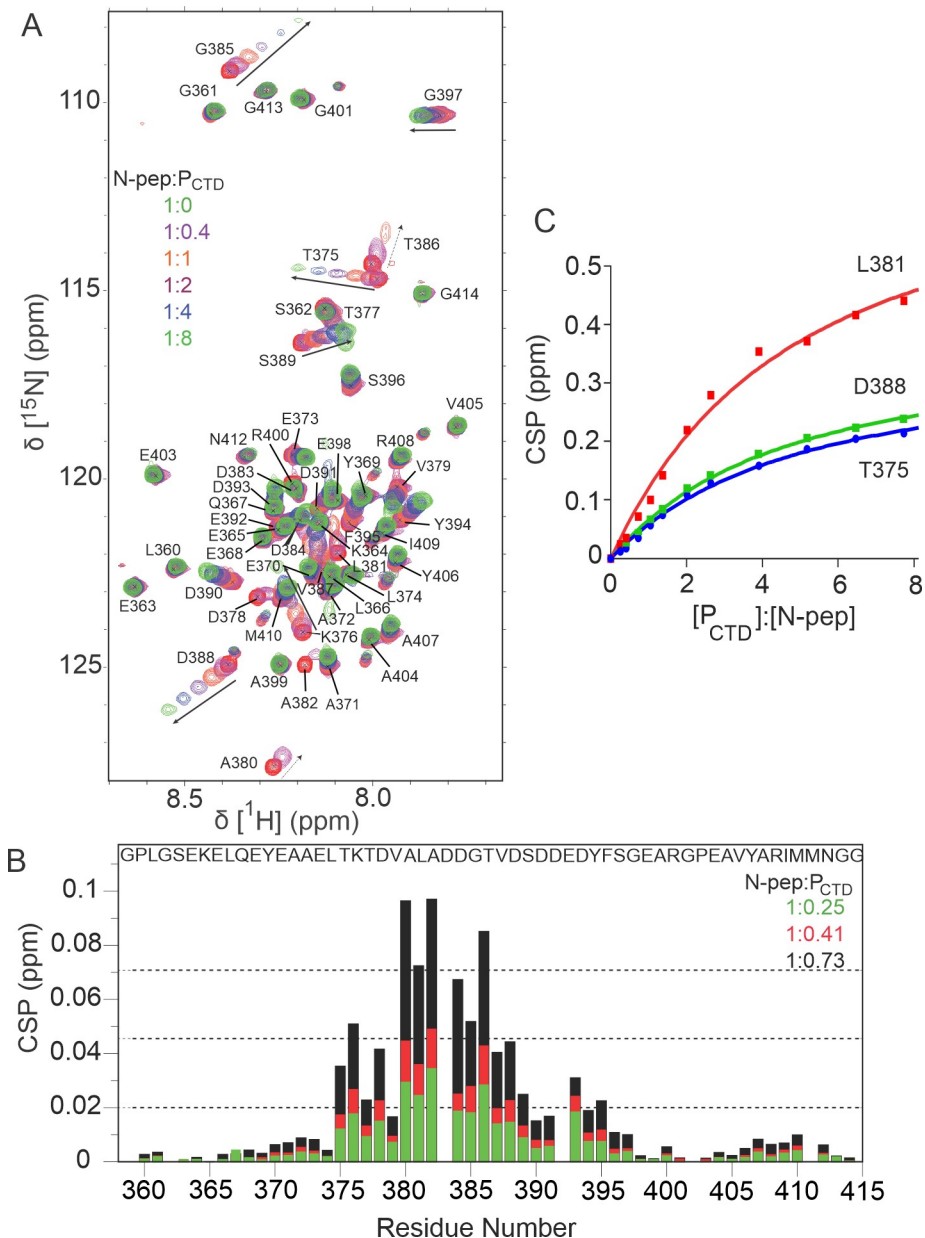

**Fig 3. Titration of ¹⁵N-labelled N-pep at 25°C and pH 6.8 with unlabelled P$_{CTD}$ up to 8-fold molar excess.** (A) ¹H-¹⁵N HSQC spectrum of ¹⁵N-labelled N-pep showing chemical shift dependence on P$_{CTD}$. Arrows indicate the shift of resonances. Signals also showing significant broadening are marked with broken arrows. (B) Plot of the change in average ¹H$^N$ and ¹⁵N chemical shift. The green, red and black colour corresponds to the chemical shift changes at N-pep: P$_{CTD}$ molar ratios of 1:0.25, 1:0.41, 1:0.73 respectively. Dashed lines indicate the mean, first standard deviation, and second standard deviation at N-pep: P$_{CTD}$ molar ratio of 1:0.73. The resonance of D383 broadens before any shifts are distinguishable. The N-pep sequence is shown. (C) Single-site saturation binding curves (K$_D$ = 224 ± 9 $\mu$M) for the residues T375, L381 and D388.

## Identification of N-pep residues important for binding P$_{CTD}$

To identify the specific residues of N that make direct contact with P$_{CTD}$, an alanine scan of the flexible loop region of N-pep was performed. Mutations included residues that shifted the most during the ¹⁵N-labelled N-pep and P$_{CTD}$ titration, the previously reported phosphorylation site S389 [42] and the negatively charged residues within the flexible loop region that

**Table 1. Kinetics data of the N-pep variants binding to $P_{CTD}$ determined by saturation binding (Xcrvfit) and two-dimensional lineshape analysis (TITAN).**

| Mutant | $K_D$ (µM) Xcrvfit[a] | $K_D$ (µM) TITAN[b] | $k_{off}$ (s⁻¹) TITAN[b] | $k_{on}$ (s⁻¹µM⁻¹) TITAN[b] |
|---|---|---|---|---|
| N-pep (WT) | 224 ± 9 | 215 ± 6 | 3164 ± 168 | 14.7 |
| K376A | 34.9 ± 2.9 | 31.7 ± 0.8 | 1803 ± 63 | 56.9 |
| D378A | 309 ± 10 | 298 ± 9 | 3977 ± 428 | 13.3 |
| A380G | 332 ± 12 | 325 ± 8 | 8943 ± 870 | 27.5 |
| L381A | 770 ± 34 | 785 ± 37 | 8870 ± 1583 | 11.3 |
| A382G | 300 ± 9 | 272 ± 8 | 7385 ± 1006 | 27.2 |
| D383A | 608 ± 46 | 603 ± 26 | 8270 ± 1427 | 13.7 |
| D384A | 546 ± 28 | 554 ± 22 | 7307 ± 1238 | 13.2 |
| T386A | 23.6 ± 1.2 | 16.2 ± 0.4 | 634 ± 15 | 39.1 |
| D388A | 323 ± 23 | 340 ± 10 | 3771 ± 215 | 11.1 |
| S389E | 87.8 ± 3.7 | 78.7 ± 2.8 | 1970 ± 102 | 25.0 |
| S389A | 98.2 ± 2.2 | 90.9 ± 1.7 | 3243 ± 191 | 35.7 |
| D390A | 100 ± 6 | 94.6 ± 1.3 | 4176 ± 184 | 44.1 |
| D391A | 160 ± 5 | 153 ± 3 | 2851 ± 160 | 18.6 |
| E392A | 180 ± 27 | 166 ± 3 | 3584 ± 184 | 21.6 |
| D393A | 87.8 ± 6.7 | 81.3 ± 4.0 | 2407 ± 379 | 29.6 |
| Cyclized N-pep (WT) | 50.6 ± 5.5 | 47.8 ± 2.4 | 724 ± 53 | 15.1 |
| Cyclized S389E | 18.6 ± 1.6 | 16.1 ± 0.6 | 554 ± 24 | 34.4 |
| Cyclized-Phospho-N-pep | 9.0 ± 0.4 | 7.4 ± 0.4 | 415 ± 23 | 56.1 |

[a]Xcrvfit: dissociation constant ($K_D$ in µM) derived from Xcrvfit software by fitting the chemical shift perturbation into a single binding site model. The chemical shift changes of at least 4 residues from one titration were used to calculate the overall $K_D$ and standard deviation.

[b]TITAN: dissociation constant ($K_D$ in µM) and off-rate ($k_{off}$ in s⁻¹) were determined by simultaneously fitting the lineshape to a two-state single binding site model using TITAN software. The on-rate ($k_{on}$ in s⁻¹ µM⁻¹) was determined by dividing the off-rate by the dissociation constant. The lineshape changes of at least 4 residues from one titration were used in TITAN to calculate the overall $K_D$ and off-rate.

could be making ionic interactions with the positive patch on $P_{CTD}$. D378, D383 and D384 on N protein are predicted to be involved in $P_{CTD}$ binding in the SAXS model but have not been experimentally investigated [14]. The variant N-pep proteins were ¹⁵N-labelled and purified, and their interactions with $P_{CTD}$ studied using ¹H-¹⁵N HSQC titration experiments, with $k_{on}$, $k_{off}$ and $K_D$ calculated using two-dimensional line-shape analysis as described above (Table 1). $K_D$ values derived from saturation binding curves were also included for comparison.

Mutating L381, D383 or D384 to alanine caused significant reductions in binding with $P_{CTD}$. The dissociation constant was at least 2.5-fold weaker compared with the wild type N-pep, due to the much more rapid dissociation. The losses of affinity suggest L381 of N-pep makes crucial hydrophobic interactions, while D383 and D384 are essential for electrostatic interactions with the positive patch on the $P_{CTD}$. The involvement of L381 indicates that in addition to the proposed electrostatic interactions [31,32], confirmed here by our mutagenic analysis, there are hydrophobic interactions between N-pep/$P_{CTD}$. Mutating D378, A380, A382 and D388 had little effect suggesting that these residues do not make essential contacts with $P_{CTD}$. Notably, the importance of L381 has not been predicted previously, and the non-critical role of D378 identified here contradicts predictions made in the model [14].

Phosphorylation of N protein has been reported to modulate viral transcription and replication [43] and is proposed to enhance the interaction between P protein and the viral nucleocapsid [44]. The phosphorylation site has been mapped to S389, with phosphorylation catalysed by cellular casein kinase II (CK-II) [42]. To investigate the effect of phosphorylation

on N/P interaction, a phosphomimetic mutant (S389E), phosphorylation-deficient mutant (S389A) and *in vitro* CK-II phosphorylated N-pep were produced and $^{15}$N-labelled (S4 Fig), before titration with unlabelled $P_{CTD}$. Phosphorylation of N-pep improved the binding affinity almost 7-fold compared with the unphosphorylated N-pep through more rapid association and extended complex lifetime, indicated by a faster on-rate and a slower off-rate (Table 1). Possibly the long-range electrostatic steering between the phosphate group and $P_{CTD}$ accelerated the complex formation after a diffusive encounter [45]. Also due to the extra negative charge introduced onto N-pep, more ionic interactions would be expected to be formed, making the complex more stable with a slower off-rate. Phosphomimetic N-pep (S389E) bound to $P_{CTD}$ 3-fold more strongly than wild type N-pep, but more weakly than phosphorylated N-pep. The kinetic differences between unphosphorylated N-pep and phosphorylated N-pep support previous mammalian cell-based studies indicating that S389 phosphorylation of N is important to the formation of the viral replication complex, enabling transcription and replication [43,44]. By analysing the interaction of the purified proteins, our data confirm the role of phosphorylation at S389 in the bimolecular interaction of P and N, distinct from any cellular interactors, compartmentalisation or phosphorylation at other sites in either protein.

Unexpectedly, the phosphorylation-deficient N-pep (S389A) also bound to $P_{CTD}$ 2-fold more tightly than unphosphorylated N-pep, with more than a 2-fold faster association rate but a similar complex lifetime. Several other alanine substitutions (K376A, T386A, D390A, D393A) were also found to bind to $P_{CTD}$ with an accelerated association rate and/or extended complex lifetime and, therefore, a tighter affinity. The reason behind the faster on-rate affected by these mutations is unclear but may be due to removal of certain steric hindrances during the complex formation process, or prevention of non-specific interactions that could slow down the diffusion and, consequently, the on-rate [46]. Alternatively, the alanine substitutions might selectively promote or stabilize a favourable transient conformation of free N-pep that enables formation of the "folded" final complex [47–50]. Apart from the faster on-rate shown by these alanine substitutions, the alanine substitution at T386 notably extended complex lifetime as indicated by the 5-fold slower off-rate, probably due to more productive hydrophobic interactions with $P_{CTD}$.

## Protein-protein docking analysis of $P_{CTD}$ and N-protein complex formation

To visualize the N-pep binding site on the $P_{CTD}$ a model of the $P_{CTD}$ and N-protein complex was generated using a combination of rigid and flexible protein docking, followed by restrained minimization. The flexible N-protein loop, which was not observed in the electron density of the RABV N protein crystal structure, was observed to dock strongly to the only polar fragment binding hotspot on the surface of $P_{CTD}$. Flexible docking generated 100 poses, which were all located near the key $P_{CTD}$ residues K211, K214, L224 and R260, consistent with our experimental observations, and binding pocket detection. These poses were filtered according to binding modes to key residues, and the best pose was subjected to further optimization. The final pose (Fig 4) maximized interactions between key residues, particularly between the N-protein residues D383 and D384 with the $P_{CTD}$ residues K211 and K214, respectively; L381 with $P_{CTD}$ L224; and phosphorylated S389 with $P_{CTD}$ R260. Finally, the flexible N-protein loop of the final pose runs antiparallel to the region K211 to K214 of $P_{CTD}$, with the interaction predominantly mediated by local hydrophobic, ionic and hydrogen bond interactions (Fig 4). Visualizing this interaction through a 20 ns molecular dynamics simulation (S1 Movie) shows that this interaction is retained.

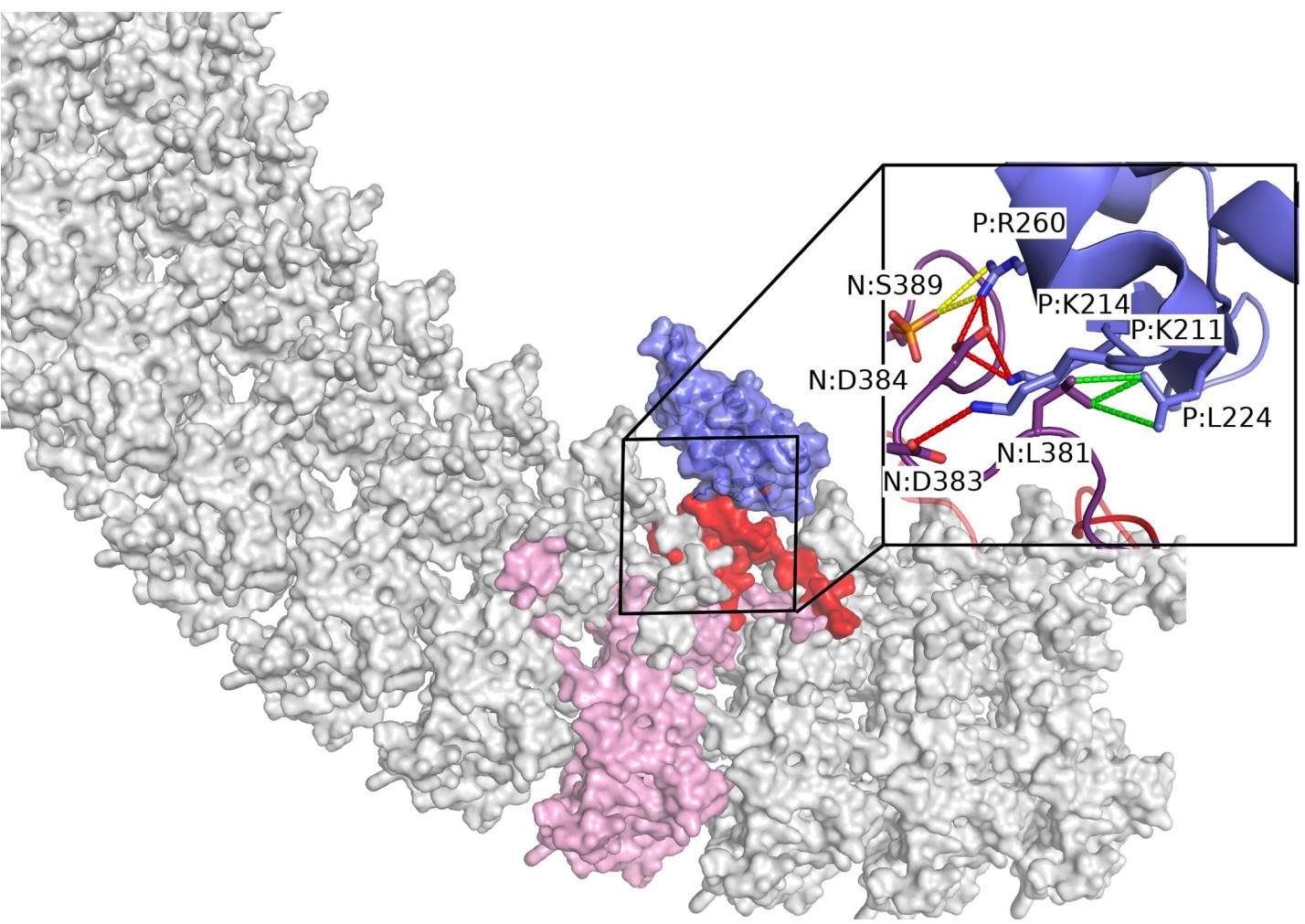

**Fig 4. Model of the complex between N protein and $P_{CTD}$.** $P_{CTD}$ (blue) preferentially docked to an exposed loop (residues 363 to 414, red) of an N protein subunit (pink) through a shallow pocket positioning the $P_{CTD}$ on the concave surface of the N protein undecamer. The inset shows the binding interface. Key N protein residues (violet) and $P_{CTD}$ residues (blue) are shown as sticks and labelled: with the inter-molecular interactions calculated using Arpeggio, including hydrophobic (green), hydrogen bond (red) and ionic (yellow), shown as dashed lines.

## $P_{CTD}$ forms tripartite complexes with pY-STAT1 and N protein

Our experimental confirmation and extension of understanding of the N binding site poses a significant question since the STAT1 and N-binding sites, while distinct, are proximal (S1 Fig). This colocalization of N and STAT1 binding to the small $P_{CTD}$ suggests that the interactions might be regulated by mechanisms such as steric hindrance, significant to the 'balance' between immune evasion and replication.

To assess the possible regulation of N protein and STAT1 interaction with P protein, we used co-immunoprecipitation (co-IP) assays of COS-7 cells co-transfected to express mCherry-fused N protein and GFP-fused $P_{CTD}$, or control proteins. IP for GFP-$P_{CTD}$ from cells co-expressing control protein (mCherry alone) indicated little to no STAT1 interaction in the absence of IFN treatment, with interaction becoming apparent following IFN treatment (Fig 5A). Thus, in common with full-length P protein [16,24], STAT1 interaction of $P_{CTD}$ alone is dependent on IFN activation. Importantly, co-expression of mCherry-N protein did not inhibit $P_{CTD}$/STAT1 interaction (Fig 5A). Similarly, the interaction of GFP-$P_{CTD}$ with

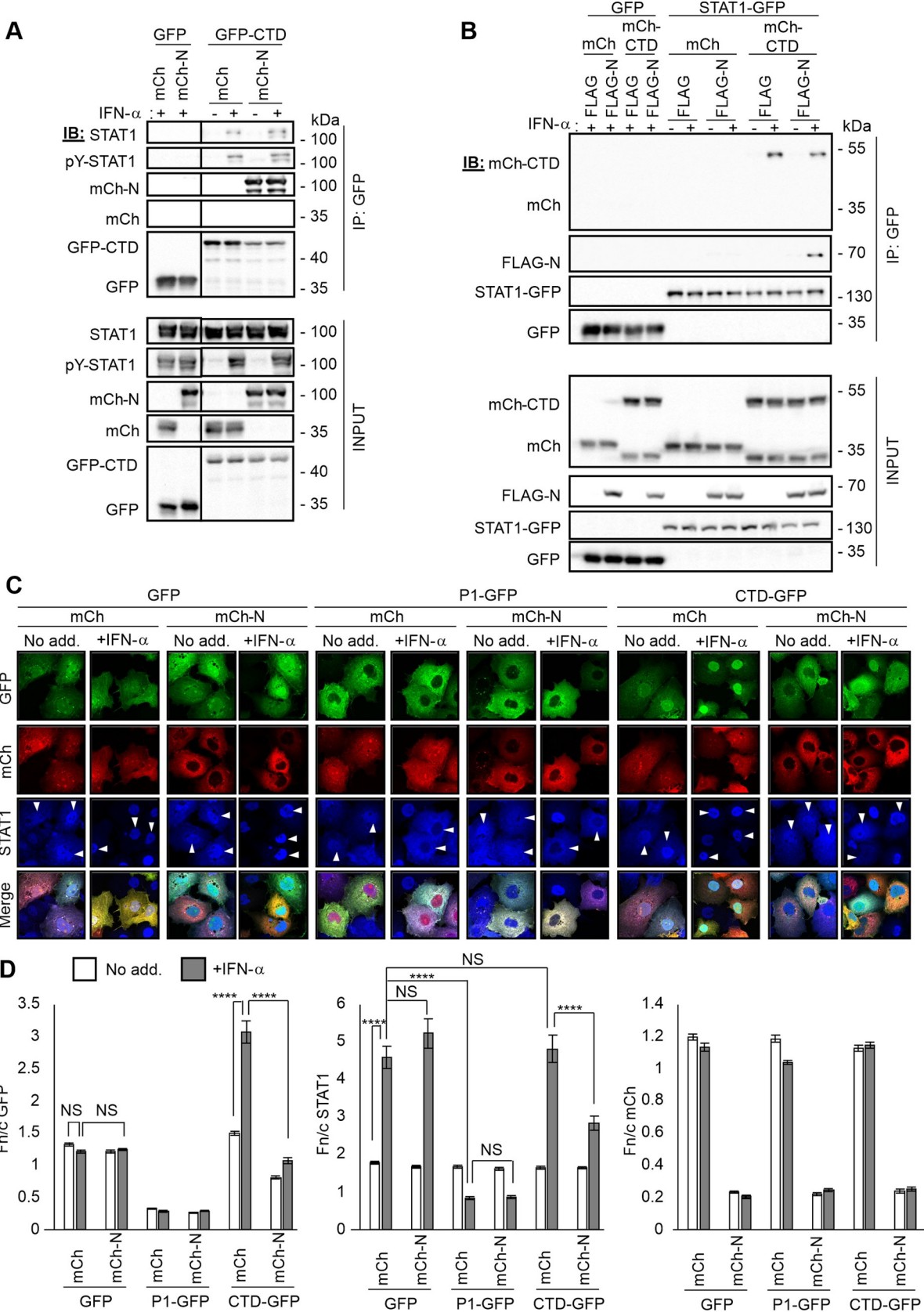

**Fig 5. STAT1, N protein and P$_{CTD}$ interact in a non-competitive fashion.** (A) COS-7 cells co-transfected to express the indicated proteins were treated 24 h post-transfection with or without IFN-$\alpha$ (1000 U/ml, 30 min) before lysis and immunoprecipitation for GFP. Lysates (input) and immunoprecipitates (IP) were analysed by immunoblotting (IB) using antibodies against the indicated proteins. Results are representative of 3 independent assays and show data from a single blot with intervening and marker lanes removed. (B) HEK-293T cells co-transfected to express the indicated proteins were treated 24 h post-transfection with or without IFN-$\alpha$ (1000 U/ml, 1 h) before lysis, immunoprecipitation for GFP and IB, as above. Results are representative of 2 independent assays. (C) COS-7 cells co-transfected to express the indicated proteins were treated 24 h post-transfection with or without IFN-$\alpha$ (1000 U/ml, 30 min) before fixation, immunofluorescent staining for STAT1 (blue) and analysis by confocal laser scanning microscopy. Representative images are shown. Arrowheads indicate cells with detectable expression of the transfected GFP- and mCherry-fused proteins. (D) Images such as those shown in (C) were analysed to calculate the Fn/c for GFP, mCherry, and immunostained STAT1 (mean ± SEM; n ≥ 43 cells for each condition; results are from a single assay representative of two independent assays). Statistical analysis used Student's $t$ test. ****, p < 0.0001; NS, not significant.

mCherry-N protein was not impaired following induction of STAT1 phosphorylation (and consequently, interaction with P$_{CTD}$) by IFN treatment (Fig 5A).

These data are consistent with a lack of steric hindrance or competitive binding to P$_{CTD}$ by N protein and STAT1, suggesting that N protein and pY-STAT1 can interact simultaneously with P$_{CTD}$. Thus, N-RNA/P replication complexes may bind to STAT1 to inhibit IFN signaling; this would provide a method to disable STAT1 signaling without requiring P protein to dissociate from replication complexes. We reasoned that if such complexes are formed, P$_{CTD}$ would bridge pY-STAT1 to N protein, which otherwise does not bind or inhibit STAT1 directly [24,51]. To assess this, we co-expressed STAT1-GFP with mCherry-P$_{CTD}$ and FLAG-N protein before treatment without or with IFN, and IP for STAT1-GFP (Fig 5B). As expected, mCherry-P$_{CTD}$ coprecipitated with STAT1 in an IFN-dependent manner in cells expressing FLAG-N or FLAG control, confirming no significant inhibitory effect of N protein on the complex (Fig 5B). Consistent with previous data (Fig 5A), N protein did not coprecipitate with STAT1 from IFN-treated or untreated cells expressing mCherry control (Fig 5B). However, co-expression of mCherry-P$_{CTD}$ conferred interaction between STAT1 and N protein, dependent on IFN (Fig 5B). Thus, N protein does not bind to STAT1 unless P$_{CTD}$ is co-expressed, by which N protein gains P$_{CTD}$-like function in binding selectively to IFN-activated STAT1, consistent with P$_{CTD}$ linking cellular N protein to pY-STAT1.

To confirm this interaction in the cellular context, we transfected COS-7 cells to express P$_{CTD}$-GFP or P1-GFP, mCherry-N protein, or controls, before treatment with or without IFN and fixation and immunofluorescence staining for STAT1. Subcellular protein localization was assessed by confocal laser scanning microscopy (CLSM) (Fig 5C) and quantitative analysis of CLSM images to derive the nuclear to cytoplasmic fluorescence ratio for each protein (Fn/c, Fig 5D) [20,24,51–53]. In untreated cells expressing controls (GFP and mCherry), STAT1 was localized between the nucleus and cytoplasm, but following IFN treatment became strongly nuclear, as expected (Fig 5C and 5D). Equivalent nuclear localization of STAT1 was observed in cells expressing mCherry-N protein despite the expected strongly cytoplasmic localization of N protein (Fig 5C and 5D), consistent with lack of interaction of N and STAT1, or STAT1-antagonistic function by N protein [24,51]. Expression of GFP fused to full-length P protein (P1, which is cytoplasmic due to the N-terminal NES (N-NES) and so results in cytoplasmic localization of the STAT1-binding P$_{CTD}$) resulted in inhibition of STAT1 nuclear localization in IFN-treated cells as expected [16,24,51], with no effect of expression of N protein (Fig 5C and 5D). P$_{CTD}$ was diffusely localized between the nucleus and cytoplasm with no inhibitory effect on STAT1 nuclear accumulation observed (Fig 5C and 5D), consistent with the lack of the strong N-NES sequence that is within the N-terminal part of P protein [22,53]. In fact, nuclear localization of P$_{CTD}$ was enhanced following IFN treatment (Fig 5C and 5D), consistent with 'piggy-backing' of cytoplasmic P$_{CTD}$ into the nucleus due to interaction with activated pY-STAT1; this is similar to effects observed for the P protein of the attenuated Ni-

CE RABV, in which the N-NES is inactivated by mutations [51]. In cells co-expressing N protein, however, $P_{CTD}$ had a much more cytoplasmic localization (Fig 5C and 5D), consistent with N/P protein interaction resulting in co-localization of a significant proportion of $P_{CTD}$ with cytoplasmic N protein. This difference also correlated with a significantly more cytoplasmic localization of IFN-activated STAT1 in cells co-expressing $P_{CTD}$ with N protein compared with cells expressing either protein alone (Fig 5C and 5D). Thus, N protein causes cytoplasmic localization of $P_{CTD}$-associated pY-STAT1; taken together with the data from the immunoprecipitation assays, this is consistent with non-competitive, simultaneous binding of $P_{CTD}$ to N protein and pY-STAT1.

In RABV-infected cells, N protein and associated RNA accumulate with P protein and L protein in Negri body compartments [13,54], which act as replication factories by concentrating and sequestering, or excluding, specific viral and cellular factors. Similar N protein-rich inclusions are observed in cells infected by other viruses of the order *Mononegavirales* such as measles, mumps, and Ebola viruses [55–57]. Sequestration of replication complexes may be important to restricting detection by cellular antiviral systems, such as through exclusion of innate immune proteins (e.g. pattern recognition receptors), although STATs and Toll-like receptors have been reported to localize with inclusion/replication compartments of certain viruses [54–59]. For N-RNA/P complexes to interact with pY-STAT1, STAT1 must be able to localize into Negri bodies and not be excluded from these complexes; consistent with this, previous data have not indicated exclusion of STAT1 from cytoplasmic regions consistent with Negri bodies (e.g. [51]). To confirm this, we infected SK-N-SH cells with RABV (MOI 5, 24 h to achieve infection of c. 100% cells) before treatment without or with IFN, and immunostaining for pY-STAT1 (S5A and S5B Fig). CLSM analysis indicated minimal pY-STAT1 signal in non-treated mock-infected cells, which increased following IFN treatment and accumulated in the nucleus. pY-STAT1 also increased following RABV infection, and was further enhanced by IFN treatment, but was excluded from the nucleus, as expected. Importantly, cytoplasmic pY-STAT1 showed no evidence of exclusion from discrete regions of the cytoplasm (S5A and S5B Fig). Co-staining of cells for N protein confirmed the formation of N-protein-enriched Negri bodies in c. 100% of cells in infected samples, and co-localization with pY-STAT1 was evident (S5C and S5D Fig). Thus, pY-STAT1 can co-localize with N-RNA/P complexes, consistent with a lack of exclusion through liquid-liquid phase compartmentalization or competitive binding to P protein by N protein, and supporting potential for interaction of STAT1 with N-RNA/P complexes (Fig 5). However, we did not observe strong sequestration of pY-STAT1 into Negri bodies, indicating that P protein-pY-STAT1 complexes formed within and outside of Negri bodies contribute to antagonism.

## Discussion

By combining NMR titrations and mutagenesis data, our study mapped the binding interfaces on N and P proteins and successfully defined key N residues involved in the interaction. This provides direct data on the interaction site of N protein with $P_{CTD}$, providing insight into the spatial relationship of key interactions involved in immune evasion and replication that are critical elements in infection and pathogenesis. The $P_{CTD}$ is a major hub of interactions by P protein, including with viral (N protein) and cellular proteins (e.g. STAT1/2/3, importins, exportins and microtubules), but precisely how these are accommodated or regulated are unknown. These data provide insights into the relationship of such interactions, providing tools for further analysis of the broader $P_{CTD}$ interactome.

As predicted [14,26,29,31,32] residues of the positive patch (K211, K214, L224 and R260) on the $P_{CTD}$ formed the key binding site for the $N_{CTD}$ loop. Importantly, our data confirmed

these residues as mediating direct interaction, discounting possible off-target effects of mutagenesis or roles of interactions with other proteins [31,32]. In the N protein, D383, D384 and phosphorylated S389 made electrostatic interactions with the positively charged residues on $P_{CTD}$, validating previous predictions [14,31] and consistent with reports that N protein phosphorylation is important for P protein-binding and transcription/replication in cellular systems [42–44]. Notably, L381 of N, which was not previously implicated in the interaction by models or experimental approaches, appears to be essential and likely makes a hydrophobic interaction with L224 of $P_{CTD}$. Furthermore, in contrast to the prediction that D378 of N protein interacts with K214 of $P_{CTD}$ [14], we found that D378 was dispensable. Consistent with critical roles in replication, all residues identified as important to the N-pep/$P_{CTD}$ interface are conserved among lyssaviruses (S6 Fig).

In contrast to predictions of the previously described model [14] that suggest an N protein binding site in the W-hole on $P_{CTD}$, the NMR titration of $^{15}$N-labelled $P_{CTD}$ with N-pep indicated only minor effects on W-hole residues on the flat face of $P_{CTD}$, compared with strong perturbations of residues close to the positive patch on the round face (Fig 2D). Analysis of the chemical shift perturbations for W265 and C271 of this W-hole is consistent with the apparent $K_D$ calculated for a single-site interaction and are likely to reflect the interaction with the positive patch. These data are in agreement with a lack of support for roles of the W-hole residues in mutagenic studies of N binding [26,32]; notably the W-hole residues, especially W265, are poorly conserved among lyssaviruses [27] (S6B Fig). The original proposal that the W-hole might be involved in $P_{CTD}$ binding [26] was based on the presence of mutations in the region corresponding to the W-hole in MOKV $P_{CTD}$ constructs (in MOKV the W is substituted for F) that were defective for N binding in a yeast two-hybrid random mutagenesis screen [32]. As all these constructs also contained mutations to the conserved positive patch, it seems likely that the lack of N binding was independent of mutations in the W-hole. While it remains possible that binding occurs at the W-hole, with an extremely weak affinity of N-pep (e.g. $K_D \gg$ 1 mM), such that the interaction is not distinguishable in the presence of the μM affinity with the positive patch, the lack of a substantial effect of specific W-hole mutations of N interaction or transcription/replication [29] suggests any such interaction is not significant. Taken together, the data indicate that the positive patch of $P_{CTD}$ is the primary N-pep binding site, and the W-hole either does not contribute (consistent with the empirical data to date) or at best contributes very weakly to the affinity.

For efficient transcription and replication, phosphorylation of S389 on N protein is required [42–44]. We found S389 phosphorylation enhances the direct N-pep/$P_{CTD}$ interaction (c. 7-fold) through more rapid association and a longer complex lifetime. However, the low micromolar affinity between CK-II phosphorylated cyclized N-pep and $P_{CTD}$ (7.4 ± 0.4 μM) is ~50-fold weaker than the reported affinity between N-RNA ring and $P_{CTD}$ (160 ± 20 nM), previously determined by SPR [14]. This discrepancy might arise from the major difference in the length of the reactant. Here, instead of the 550 kDa $N_{10}$-RNA ring used previously for SPR, a 57-residue segment of the $N_{CTD}$ flexible loop region was used for NMR. This might not comprise all components of the N protein-$P_{CTD}$-binding interface in the properly folded form. If the W-hole plays a role in N-RNA/P interaction [14], with a very weak affinity, in the context of full-length N protein and as an array of binding sites in the $N_{10}$-RNA ring, the avidity would lead to a much tighter interaction as well.

The affinity between N-pep and $P_{CTD}$ was made significantly stronger through cyclization. The idea of a conformational change and folding upon the $N_{CTD}$ binding with $P_{CTD}$ [60] is consistent with previous analysis showing that a monoclonal antibody against N bound exclusively in the context of an N-RNA complex with P, with the epitope mapped to the $N_{CTD}$ loop region [61]. The antibody did not recognize the epitope, mapped to residues 388–407, when P

was removed from the complex, or in RNA-free $N^0$ protein, which binds to the N-terminal region (residues 23–50) of P protein, such that P protein provides a chaperone function for newly synthesized N protein prior to encapsidation of the viral RNA [61].

Other than N-RNA, P protein interacts with multiple cellular proteins with interaction with STAT1 shown to be highly significant to infection and pathogenesis through enabling suppression of antiviral STAT1 signaling *via* a critical binding site within the $P_{CTD}$. The data here, and in our previous study [16] place the N-RNA and STAT1 binding sites in the $P_{CTD}$ in close proximity on the round face of the CTD; one potential outcome of this would be that binding might be competitive such that P protein would engage either in replication or, following STAT1 activation, be diverted to immune evasion. Our data do not support this model and, in fact, indicate that $P_{CTD}$ can simultaneously bind to N-RNA and STAT1. This is consistent with the fact that the sites are proximal but do not overlap; furthermore, the finding that mutations disabling the STAT1-binding site do not prevent N-binding or replication are indicative of distinct requirements of the respective binding interfaces [16]. In agreement with the apparently non-competitive binding, N protein, that does not have apparent STAT1 binding/antagonistic activity alone, can bind to STAT1 when in complex with $P_{CTD}$. Thus, it appears that N-RNA/P protein complexes do not need to be disrupted to enable immune evasion. Since full-length P protein forms dimers which are predicted to interact with N-RNA *via* the $P_{CTD}$ of only one of the monomers [14], it is possible that STAT1 binding to P protein will involve the free $P_{CTD}$, although our data indicate that the bound $P_{CTD}$ is also able to form interactions. Whichever model is correct awaits further structural definition, but clearly the data indicate that P protein-STAT1 interaction for immune evasion does not interfere with interactions of P protein with N-RNA critical for replication, or *vice versa*; indeed, pY-STAT1 was not excluded from Negri bodies in infected cells, consistent with a lack of competitive binding. Together these data suggest that replication complexes may participate directly in the antagonism of STAT1. This structural organization of the $P_{CTD}$ would therefore appear to provide a novel and efficient interaction hub at the interface of active replication and antagonism of the resulting antiviral response. It is also possible that the capacity of N-RNA/P to interact with STAT1 results in recruitment of STAT1 with some positive role in viral replication. Consistent with this, STATs can have ambiguous (pro- or antiviral) roles in infection [62]. Delineation of these possibilities is the focus of further research.

## Materials and methods

### Constructs and mutagenesis

N-pep (residues 363–414, comprising the $N_{CTD}$ loop and flanking helices α13 and α14) was cloned into a pGEX-6P-3 vector using *BamHI-XhoI* restriction sites, generating a construct to express N-pep with an N-terminal GST-tag followed by a PreScission protease cleavage site. $P_{CTD}$ (residues 186–297) was cloned into pET28a vector using *NdeI-EcoRI* restriction sites, producing a construct to express $P_{CTD}$ as a His-tagged protein, with a TEV cleavage site between the His-tag and $P_{CTD}$. The last residue of $P_{CTD}$ C297 was mutated to serine to avoid the formation of an intermolecular disulfide bond.

Mutations were introduced into the N-pep or $P_{CTD}$ constructs using PrimeSTAR Max DNA Polymerase (Takara) according to the manufacturer's instructions, and using mutagenic primers designed according to Zheng et al. [63]. The amplified material was digested with DpnI for 1.5 hours at 37°C to remove wild type bacterially expressed plasmid before transformation into Top10 *E. coli* cells. Mutation of the plasmid was confirmed by sequencing at the Micromon sequencing facility (Monash University).

Mammalian expression constructs to express CTD fused to the C- or N-terminus of GFP (GFP-$P_{CTD}$ (P protein residues 174–297) and $P_{CTD}$-GFP (residues 173–297) respectively) and N protein fused to the C-terminus of mCherry (mCherry-N protein) were generated using pEGFP-C1, pEGFP-N3 or pmCherry-C1 plasmids, and have been described previously [23,29,64]. Plasmids were generated in this study by standard restriction ligation. The plasmid for expression of FLAG-N protein was generated by insertion of 3xFLAG sequences into pΔEGFP (pEGFP-C1 deleted for GFP) [23] followed by insertion of N protein in frame C-terminal to the 3xFLAG sequence. Plasmid for expression of P1-GFP was generated by insertion of full-length P1 sequence in frame N-terminal to GFP in pEGFP-N3, and plasmid for mCherry-$P_{CTD}$ was generated by insertion of $P_{CTD}$ C-terminal to mCherry in pmCherry-C1.

## Protein expression and purification

Unlabelled N-pep and $P_{CTD}$ protein samples were expressed in *E. coli* BL21(DE3) in 2YT auto-induction media (N-5052) in which the expression of the target protein is induced upon the change of glucose-lactose metabolic state during the culture growth [65]. After $OD_{600}$ reached 0.6~0.8 at 37˚C, the culture was transferred to 16˚C for overnight incubation. To produce uniformly $^{15}$N-labelled N-pep and $P_{CTD}$ protein samples for NMR, cells were grown in N-5052 supplemented with 1 g/L of $^{15}$NH$_4$Cl as a sole source of nitrogen [65]. To label N-pep with $^{13}$C and $^{15}$N isotopes, cells were grown in 2YT media until $OD_{600}$ reached 0.6~0.8 at 37˚C, then pelleted and resuspended in a quarter the original culture volume of fresh N-5052 media supplemented with 2 g/L of $^{15}$NH$_4$Cl (Sigma-Aldrich) and 4 g/L of D-[$^{13}$C] glucose (Sigma-Aldrich) as sole sources of nitrogen and carbon [66]. The resuspended cell culture was transferred to 16˚C and induced with 0.25 mM isopropyl β-D-1-thiogalactopyranoside after 1-hour incubation at 16˚C, harvested after a 15-hour induced period.

For purification of N-pep, bacterial pellets were resuspended in 10 mM Na$_2$HPO$_4$, 2 mM KH$_2$PO$_4$, 137 mM NaCl and 3 mM KCl (pH 7.4) and a protease inhibitor cocktail tablet (Roche). Cells were lysed using an Avestin EmulsiFlex-C3 homogenizer and the debris removed by centrifugation at 13,000 g, 4˚C for 45 minutes. The supernatant was applied to Glutathione Sepharose 4B resin (GE Healthcare), before washing with Phosphate-buffered saline (PBS) and incubation with Tris-buffered saline (TBS) solution containing PreScission protease (200 μL of 3 mg/ml purified PreScission protease in 25 mL of TBS) for 2 hours at 4˚C to remove the GST-affinity tag while bound to the Glutathione Sepharose matrix. The cleaved N-pep was washed out of the column with TBS buffer and concentrated using an Amicon Ultra 3 kDa centrifugation filter (Millipore). N-pep was then further purified by reversed-phase high performance liquid chromatography (RP-HPLC) using a 0 to 60% acetonitrile gradient (0.1% trifluoroacetic acid) applied over 60 min at a flow rate of 5 ml/min using a C18 column (Agilent ZORBAX 300SB-C18, 5 μm, 9.4×250 mm). To generate cyclic N-pep variants, the cleaved linear precursor was concentrated and buffer-exchanged into 100 mM NH$_4$HCO$_3$ (pH 8.5) and oxidized overnight at room temperature before purification by RP-HPLC. The fractions collected from RP-HPLC were analysed by mass spectrometry to confirm the molecular weight, and the oxidation states for cyclic N-pep, before freeze drying. Typically, 3 to 4 mg of N-pep were purified from 1 L of bacterial culture.

The purification of $P_{CTD}$ was performed as described previously [67]. Briefly, recombinant $P_{CTD}$ was purified by TALON metal affinity chromatography (Clontech), and the His-tag was removed by overnight TEV treatment (0.5 ml of 1.8 mg/ml purified TEV per 50 ml of protein sample). Protein samples were further purified by size-exclusion chromatography using a HiLoad 16/60 Superdex 75 column (GE Healthcare) in 50 mM Na$_2$HPO$_4$/NaH$_2$PO$_4$ (pH 6.8), 100 mM NaCl.

## NMR experiments and resonance assignments

All NMR data were collected at 25°C using a Bruker 700 MHz AVANCE III HD or a Bruker 800 MHz AVANCE II spectrometer, both are equipped with cryogenically cooled triple resonance probes to maximize sensitivity. Spectra were processed using *NMRPipe* [68] in which data were essentially Fourier-transformed after being multiplied by a Lorentz–Gaussian function in the direct dimension and cosine bells in the indirect dimensions and zero-filled once. Spectra were analysed using *NMRFAM-SPARKY* [69].

The spectra assignment of N-pep or its variant was performed using a 200 μM $^{13}$C,$^{15}$N-labelled sample in 50 mM $Na_2HPO_4$/$NaH_2PO_4$ (pH 6.8) and 100 mM NaCl. A set of five triple resonance spectra were acquired: 3D HNCO, HN(CA)CO, HNCACB, HN(CO)CACB and CC(CO)NH. All the triple resonance spectra were recorded using 10% non-uniform sampling (NUS) [70] with multidimensional Poisson gap scheduling [71], and reconstructed using *qMDD* [72] with the compressed sensing algorithm before being processed in *NMRPipe* [68]. Assignments were done by manually verifying and supplementing the initial auto-assignment generated by the *PINE* server [73]. $^{13}$Cα and $^{13}$Cβ chemical shifts were referenced using the random coil chemical shifts and secondary structure chemical shifts from RefDB [74], and were used to calculate secondary structure propensities using *SSP* [75]. Figures were generated using *PyMOL* [76] and *ESPript* [77]. Sequence alignment was performed with *Clustal X* [78] and the secondary structure elements were added using *DSSP* [79].

## NMR titration experiments

The binding of N-pep variants to $P_{CTD}$ was monitored using $^1$H-$^{15}$N HSQC titration experiments. Both N-pep and $P_{CTD}$ samples were dialyzed overnight against the same buffer with 50 mM $Na_2HPO_4$/$NaH_2PO_4$ (pH 6.8) and 100 mM NaCl using the mini dialysis kit with 1 kDa cut-off (Amersham Biosciences). Concentrated unlabeled $P_{CTD}$ was gradually added into the $^{15}$N-labelled N-pep variant sample until reaching at least 5-fold molar excess. The $^1$H-$^{15}$N HSQC spectra (2048×256 data points) were recorded at apo and at least six different titration points. The reverse titration of $^{15}$N-labelled $P_{CTD}$ with unlabelled N-pep (wild type) was conducted in a similar manner. Seven spectra were recorded at $P_{CTD}$: N-pep molar ratios of 1:0, 1:0.4, 1:0.8, 1:1.5, 1:3, 1:6 and 1:9.

Resonances showing obvious chemical shift changes yet remaining well-resolved were selected to calculate binding affinity by fitting the titration curve into a model for a protein with only one ligand binding site using *Xcrvfit* software [80]. Dilution through the titration was taken into account. The chemical shift perturbation (CSP) of each residue was calculated by the formula [81]:

$$CSP = \sqrt{\Delta\delta_H^2 + 0.154^2 \cdot \Delta\delta_N^2}$$
Eq 1

For all $^{15}$N-labelled N-pep variants with $P_{CTD}$ NMR titration data, two-dimensional lineshape analysis was done using the software *TITAN* [41]. The off-rate ($k_{off}$) and dissociation constant ($K_D$) were determined by simultaneously fitting the lineshape of selected shifted yet resolved peaks to a two-state single binding site model. The on-rate ($k_{on}$) was determined by the equation:

$$k_{on} = \frac{k_{off}}{K_D}$$
Eq 2

## Cell culture and transfection

COS-7 and HEK-293T cells were maintained in DMEM supplemented with 10% FCS and Gluta-MAX (Life Technologies), 5% $CO_2$, 37°C. SK-N-SH cells were maintained in EMEM

supplemented with 10% FCS. Plasmid transfections used Lipofectamine 2000 (Invitrogen) according to the manufacturer's instructions.

## Co-immunoprecipitation (Co-IP) and immunoblotting (IB) assays

Co-IPs were performed using COS-7 cells or HEK-293T cells growing in 6-cm tissue culture plates. Cells were transfected to express the indicated proteins and treated without or with 1000 U/mL recombinant human IFN-α (PBL Assay Science, Cat#PBL-11200-2) for 0.5 h (analysis of endogenous STAT1) or 1 h (analysis of STAT1-GFP) before lysis using lysis buffer (10 mM Tris/Cl pH 7.5; 150 mM NaCl; 0.5 mM EDTA; 0.5% NP-40, 1 × Protease Inhibitor Cocktail (PIC; Sigma-Aldrich Cat #11697498001) and 1x PhosSTOP) for 30 min at 4˚C. Supernatants were collected by centrifugation at 12,000 g for 10 min at 4˚C and 10% of the cleared lysate was collected for 'input' analysis; the remaining lysate was subjected to IP using 10 μL of GFP-Trap beads (Chromotek). Beads were washed 3 times with dilution buffer (10 mM Tris/Cl pH 7.5; 150 mM NaCl; 0.5 mM EDTA, 1 × PIC, 1x PhosSTOP). Protein for IB analysis was eluted by resuspension of beads in SDS-PAGE sample loading buffer and incubation at 90˚C for 10 min. The lysate (input) and IP samples were separated by SDS-PAGE before analysis by western blot using antibodies for GFP (Roche Applied Science, catalog no. 11814460001), mCherry (Abnova, catalog no. PAB18013 or Abcam, catalog no. ab167453), FLAG (Sigma-Aldrich, catalog no. F1804), STAT1 (BD Biosciences, catalog no. 610185 or Cell Signaling Technology, catalog no. 14994) or pY-STAT1 (Cell Signaling Technology, catalog no. 9176).

## Confocal laser scanning microscopy (CLSM) analysis

For analysis of transfected cells, COS-7 cells growing on coverslips were co-transfected with plasmids to express GFP-fused P or $P_{CTD}$, mCherry-fused N, or controls, using Lipofectamine 2000 (ThermoFisher) according to the manufacturer's instructions. 24 h post-transfection, cells were treated without or with IFN-α (1000U/ml) for 30 min prior to fixation with 3.7% formaldehyde (10 min) and permeabilization with 90% methanol (5 min). Cells were then immunostained with anti-STAT1 antibody (CST, Cat# 14994; 1:1000, overnight, 4˚C) followed by Alexa Fluor-647 conjugated secondary antibody (ThermoFisher, Cat# A-21244; 1:1000, 90 min, RT). Coverslips were mounted onto glass slides using Mowiol mounting solution. Cells were imaged by CLSM using a Nikon C1 Inverted confocal microscope with 60× oil immersion objective.

For analysis of infected cells, SK-N-SH cells growing on an 8-well chamber slide (Corning, NY, USA) were infected with RABV CE(NiP) strain [51] at a MOI of 5. At 24 hpi, cells were treated with or without human IFN-α (1000U/ml, 30 min) and fixed in 4% paraformaldehyde (30 min) before permeabilization with 90% methanol as above. After blocking with Blocking-One P (Naclai tesque, Kyoto, Japan, 30 min), cells were incubated with anti-pY-STAT1 (CST #9167, 1:1000), with or without anti-RABV N protein mouse monoclonal antibody [82] (anti-N mAb #13–27, 1:30,000) at 25˚C for 2 h, followed by Alexa Fluor 594-conjugated donkey anti-mouse IgG (1:1000), Alexa Fluor 488-conjugated donkey anti-rabbit IgG (1:1000) and Hoechst 33342 (25˚C, 1 h). Cells were then mounted using ProLong Diamond Antifade Mountant (Invitrogen) and imaged by CLSM using an LSM 710 confocal microscope system (Carl Zeiss, Jena, Germany).

## $P_{CTD}$/N-protein complex homology modelling

The interaction between $P_{CTD}$ and N-protein was modelled in Maestro (Schrodinger suites) through flexible peptide docking, restrained rigid protein-protein docking, and restrained minimization. Prior to docking, the addition of the phosphoryl group to N-protein residue

S389 was carried out using Maestro, to reflect biological conditions of binding. A homology model of the *Nishigahara* N-protein monomer and the oligomeric trunk were built using the cryo-EM structure of vesicular stomatitis virus N-protein [36] (PDB ID: 2WYY) using Modeller [83] and MacroModel (Schrodinger). The experimental crystal structure of the $P_{CTD}$ (PDB ID: 1VYI) and the homology model of N-protein were minimized in Maestro using the OPLS3e forcefield (Prime). The top poses from rigid and flexible docking of the proteins in Maestro were minimized and analyzed using Arpeggio [84]. The final pose was selected based on the docking energy, and satisfaction of experimental restraints.

## Molecular dynamics simulations

The $P_{CTD}$-N-protein complex was capped with acetyl and N-methyl groups at the N-, and C-termini. The forcefield of the phosphorylated S389 (N-protein) was taken from http://research.bmh.manchester.ac.uk/bryce/amber/. The topology of the phosphorylated peptide of N-protein was generated in ambertool20 [85] and converted into Gromacs format using acpype [86].

All the MD simulations were performed using GROMACS (version 2020). We applied Amber ff99SB-ILDN [87] force field and TIP3P water model [88] for the system. The complex was solvated in a periodic cubic box where its wall was 1 nm away from the complex atoms. The net charge of the system was neutralized by $Na^+$ ions. All bonds were constrained with the LINCS algorithm [89]. The Verlet algorithm was used to generate the dynamics of the system at a timestep of 2 fs. The long-range electrostatic interactions were computed by the Particle Mesh Ewald method [90]. The temperature was maintained at 300 K using a Berendsen coupling [91] with a coupling time of 0.1 ps while the pressure was controlled at 1 atm by a Parrinello–Rahman barostat [92] with a pressure relaxation time of 2 ps. The cut-off distances for both van der Waals interactions and the short-range neighbour list were 1.2 nm. Energy minimization was carried out for 50000 steps of the steepest descent algorithm. The system was then equilibrated over 100 ps at constant volume and followed by the ensemble at a constant pressure of 1 atm for 100 ps. Weak harmonic positional restraints on the complex atoms with a force constant of 1000 kJ $mol^{-1}nm^{-2}$ were imposed during the minimization and these initial equilibration steps. Subsequent production runs were carried out for 20 ns at 300 K in the NPT ensemble in triplicate without the constraints of all the $P_{CTD}$-N-protein atoms except for the interacted pairs. The movie was generated using VMD [93].

## Supporting information

**S1 Fig. Surface image of the $P_{CTD}$.** In red is the location of the STAT1 binding site (I201 to F209, D235 to I237) which is proximal to the positive patch (K211, K214, R260) and the hydrophobic residue L224 that bind the flexible peptide of the N protein ($N_{pep}$) and characterized in this study, and shown in blue. The residues of the W-hole (W265 and M287) are on the opposite side of the protein and have been suggested as an additional N protein binding site. (TIF)

**S2 Fig. Overlay of $^{15}N$ HSQC of titration of $^{15}N$-labelled $P_{CTD}$ with N-peptide.** Titration of 100 μM $P_{CTD}$ with linear wild-type N-pep, 0 (red), 50 (magenta), 100 (yellow), 200 (green), 400 (pink), 800 (tomato), 1200 (blue) μM. Spectra were recorded at 25°C and pH 6.8. (TIF)

**S3 Fig. Lineshape analysis performed on wild type $^{15}N$-labelled N-pep with $P_{CTD}$ titration data.** The lineshape of G397 (grey) of N-pep at apo state and 11 different titration points (5 shown here) were fitted into a two-state binding model (magenta) with a $K_D$ of 215 ± 6 μM

and $k_{off}$ of 3164 ± 168 s$^{-1}$.
(PDF)

**S4 Fig. An overlay of $^1$H-$^{15}$N HSQC spectra of Ser389-phosphorylated cyclized N-pep (red) and non-phosphorylated form (green).** Spectra were recorded at 25°C and pH 6.8. Ser389 and phosphorylated S389 (pS389) were labelled in the spectra.
(TIF)

**S5 Fig. STAT1 can co-localize with Negri bodies.** SK-N-SH cells were infected with CE-NiP RABV (MOI = 5, 24 h) before fixation and immunostaining for pY-STAT1 (A, B) or for both N protein and pY-STAT1 (C, D); Hoechst 33342 was used to localize nuclei (DNA). Regions within the dashed white box (A, C) are expanded in B and D.
(TIF)

**S6 Fig.** Multiple sequence alignment of Lyssavirus N (A) and P proteins (B). The residues critical for N-pep/PCTD interaction are marked with a green dot. Although P protein is much less conserved between lyssaviruses than N, the critical residues identified within the N/P interface appear to be highly conserved among the lyssaviruses. Protein sequences were retrieved from the GenBank: RABV (N: O55611, P: Q9IPJ8), ABLV (N: Q8JTH3, P: Q8JTH2), EBLV1 (N: A4UHP8, P: A4UHP9), EBLV2 (N: A4UHQ3, P: A4UHQ4), DUVV (N: Q66453, P: O56774), IRKV (N: Q5VKP6, P: Q5VKP5), ARAV (N: Q6X1D8, P: Q6X1D7), KHUV (N: Q6X1D4, P: Q6X1D3), MOKV (N: P0C570, P: P0C569), LBV (N: Q82994, P: O56773), WBCV (N: Q5VKP2, P: Q5VKP1); accession numbers are shown in parentheses.
(DOCX)

**S1 Movie. Molecular dynamics simulation of interaction between the P$_{CTD}$ and N protein.**
(MPG)

## Acknowledgments

We acknowledge Cassandra David for assistance with tissue culture, and the facilities and technical assistance of the NMR facility (University of Melbourne) and the Monash Micro Imaging facility (Monash University).

## Author Contributions

**Conceptualization:** Jingyu Zhan, Angela R. Harrison, Gregory W. Moseley, Paul R. Gooley.

**Data curation:** Jingyu Zhan, Angela R. Harrison, Gregory W. Moseley, Paul R. Gooley.

**Formal analysis:** Jingyu Zhan, Angela R. Harrison, David B. Ascher, Gregory W. Moseley, Paul R. Gooley.

**Funding acquisition:** David B. Ascher, Gregory W. Moseley, Paul R. Gooley.

**Investigation:** Jingyu Zhan, Angela R. Harrison, Stephanie Portelli, Thanh Binh Nguyen, Isshu Kojima, Siqiong Zheng, Fei Yan, Tatsunori Masatani, Stephen M. Rawlinson, Ashish Sethi.

**Methodology:** Jingyu Zhan, Angela R. Harrison, Stephanie Portelli, Thanh Binh Nguyen, Isshu Kojima, Siqiong Zheng, Fei Yan, Tatsunori Masatani, Stephen M. Rawlinson, Ashish Sethi, Naoto Ito, David B. Ascher, Gregory W. Moseley, Paul R. Gooley.

**Project administration:** Paul R. Gooley.

**Resources:** Naoto Ito, David B. Ascher, Gregory W. Moseley, Paul R. Gooley.

**Supervision:** Naoto Ito, David B. Ascher, Gregory W. Moseley, Paul R. Gooley.

**Validation:** Jingyu Zhan, Angela R. Harrison, David B. Ascher, Gregory W. Moseley, Paul R. Gooley.

**Visualization:** Jingyu Zhan, Angela R. Harrison, David B. Ascher, Gregory W. Moseley, Paul R. Gooley.

**Writing – original draft:** Jingyu Zhan, Angela R. Harrison, Gregory W. Moseley, Paul R. Gooley.

**Writing – review & editing:** Jingyu Zhan, Angela R. Harrison, Stephanie Portelli, Thanh Binh Nguyen, Isshu Kojima, Siqiong Zheng, Fei Yan, Tatsunori Masatani, Stephen M. Rawlinson, Ashish Sethi, Naoto Ito, David B. Ascher, Gregory W. Moseley, Paul R. Gooley.

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
