## [Decision Letter · Decision Letter 0]

29 Mar 2021

Dear Professor Gooley,

Thank you very much for submitting your manuscript "Definition of the immune evasion-replication interface of rabies virus P protein" for consideration at PLOS Pathogens. As with all papers reviewed by the journal, your manuscript was reviewed by members of the editorial board and by several independent reviewers. In light of the reviews (below this email), we would like to invite the resubmission of a significantly-revised version that takes into account the reviewers' comments.

It would be best if you addressed the major concern of reviewer one listed under 1). 

The major concern of reviewer 2 "One major concern is that the homology modeling and molecular dynamics simulation were performed with undecamer of N, which is an artificial oligomer." Needs also be addressed. The concerns of both reviewers seem to requirer new experiments.

We cannot make any decision about publication until we have seen the revised manuscript and your response to the reviewers' comments. Your revised manuscript is also likely to be sent to reviewers for further evaluation.

Sincerely,

Matthias Johannes Schnell, PhD

Associate Editor

PLOS Pathogens

Ana Fernandez-Sesma

Section Editor

PLOS Pathogens

Kasturi Haldar

Editor-in-Chief

PLOS Pathogens

orcid.org/0000-0001-5065-158X

Michael Malim

Editor-in-Chief

PLOS Pathogens

orcid.org/0000-0002-7699-2064

Reviewer's Responses to Questions

**Part I - Summary**

Reviewer #1: In their manuscript entitled « Definition of the immune evasion-replication interface of rabies virus (RABV) P protein”, Zhan and colleagues have characterized the interaction between the Cterminal domain of RABV P (PCTD, residues 186 to 297) and RABV N protein.

The PCTD was previously proposed to bind a flexible loop of N protein that is not visible in crystal structures. They have characterized the interaction between PCTD and N-pep (residues 363-414), (both expressed in E. Coli). Using NMR, and molecular modelling they show that N protein residues, Leu381, Asp383, Asp384 and phospho-Ser389, are likely to bind a single site of the PCTD formed by a positive patch made of Lys211, Lys214 and Arg260. As this site is near the previously defined STAT1 binding site on PCTD, they have characterized the ability of N and STAT1 to bind simultaneously the PCTD and show that STAT1 and N do not compete to bind P protein.

The results are convincing and are interesting for those who worked in the field. Therefore, those data deserve to be published.

However, I have some remarks that need to be addressed before publication as they may modify the conclusions of the article.

Reviewer #2: Like other members of the Rhabdoviridae, rabies virus has a characteristic bullet shape virion with thousands of its N protein stringed together by its single, negative-strand RNA genome into the nucleoprotein capsid within a glycoprotein-decorated envelope. Structures for most viral proteins (or their homologous proteins in other members of the same family) are known except for its small phosphoprotein (P). P is known to use its C-terminal domain (Pctd) to mediate attachment of viral RNA polymerase (L) to N at the inner wall of the capsid and also to play roles in evading host immune response by binding to STAT1 during infection. But the details of interactions of P with N are not known, though structure of the Pctd and STAT1 is available. In the current study, Harrison et al. combined NMR titrations, homology modeling, molecular dynamics simulation, mutagenesis, and cell-based assays with fluorescent microscopy to map the binding interfaces on N and P proteins and define key N residues involved in the interaction. Surprisingly, the mapped interaction site is proximal to the defined STAT1 binding site on P, yet their fluorescent data indicated that STAT1 and N protein did not compete for P protein. The P proteins of rabies and related viruses in the Rhabdoviridae are recalcitrant to direct structural studies to date. To my knowledge, the data presented in the current study represent the first observation of the interaction site of N protein with Pctd. Though a direct observation of P, N and STAT1 at high resolution is still needed to reconcile with the above fluorescent data, the results as presented should already be of general interests for those working on rabies virus or related viruses (notably VSV, which is a popular tool for engineering for vaccine and cancer vectors). The paper is clearly written and methods described in great details.

**Part II – Major Issues: Key Experiments Required for Acceptance**

Reviewer #1: 1) In the abstract, the authors conclude: “it appears that interactions critical to replication and immune evasion can occur simultaneously with the same molecules of P protein and propose that replication complexes might be directly involved in STAT1 antagonism”. A similar idea is developed in the discussion (after line 569).

I disagree with this idea: N and the ribonucloproteins are essentially present inside the Negri bodies (Nikolic et al. Nat Com. Ref 13) whereas P shuttles between the cytoplasm and the Negri bodies. In the transfection system that the authors have used, the Negri bodies which are the viral factories are not present and there is no viro-induced compartmentalization. This viro-induced compartmentalization, in which genome transcription and replication takes place, is mediated by liquid-liquid phase separation. STAT1 might be excluded from the viral factory (or poorly present in it) (for a discussion on the interplay between viral factories and innate immunity, see the review by Nevers et al. Biochim Biophys Acta Mol Cell Res. 2020 Dec;1867(12):118831). Therefore, it is highly plausible that STAT 1 and N never colocalize in infected cells and thus do not simultaneaously interact with the same P protein.

Therefore, either the authors demonstrate the presence of STAT 1 in the viral factories (i.e. they perform experiments in infected cells) or they discuss the alternative hypothesis at least at the same level as theirs.

2) It is a bit disappointing that the mutants (with the exception of S389E) have only been constructed in the context of N-pep and not in the context of its cyclized version, as the latter is clearly more relevant. Nevertheless, I understand that it can take a long time to produce and characterize all the mutants in the cyclized version.

Reviewer #2: One major concern is that the homology modeling and molecular dynamics simulation were performed with undecamer of N, which is an artificial oligomer. In the virion, N is known to begin with a decameric ring of N, with gradual increasing radius, reaching a final helical parameter of about 38-39 N/turn in the trunk part of the bullet (Ge et al., Science 2010). Though the location of L attachment by P in the virion is not defined as yet, it can be ruled out to be near the tip (the decameric ring) part of the bullet. The orientation of N is drastically different in the undecamer ring as compared to N in the trunk part of the virion. The authors should repeat these computational work with N oligomers properly oriented as in the trunk, perhaps with a box containing 4x4 N subunits.

**Part III – Minor Issues: Editorial and Data Presentation Modifications**

Reviewer #1: The reference 47 (Green and Luo, PNAS) is mentioned in the list of references (it is indeed a pertinent one). However, I cannot find where it is quoted in the text.

Reviewer #2: (No Response)

PLOS authors have the option to publish the peer review history of their article (what does this mean?). If published, this will include your full peer review and any attached files.

Reviewer #1: No

Reviewer #2: No
---

## [Decision Letter · Decision Letter 1]

18 Jun 2021

Dear Professor Gooley,

We are pleased to inform you that your manuscript 'Definition of the immune evasion-replication interface of rabies virus P protein' has been provisionally accepted for publication in PLOS Pathogens.

Best regards,

Matthias Johannes Schnell, PhD

Associate Editor

PLOS Pathogens

Ana Fernandez-Sesma

Section Editor

PLOS Pathogens

Kasturi Haldar

Editor-in-Chief

PLOS Pathogens

orcid.org/0000-0001-5065-158X

Michael Malim

Editor-in-Chief

PLOS Pathogens

orcid.org/0000-0002-7699-2064

Reviewer Comments (if any, and for reference):

Reviewer's Responses to Questions

**Part I - Summary**

Reviewer #1: This is a revised version of the manuscript entitled « Definition of the immune evasion-replication interface of rabies virus (RABV) P protein” in which Zhan and colleagues have characterized the interaction between the Cterminal domain of RABV P (PCTD, residues 186 to 297) and RABV N protein.

My main concern was about a possible exclusion of STAT1 from the Negri bodies where P and N accumulate.

The authors show in their supplementary figure 5 that this is not the case. STAT 1 is indeed present in the Negri bodies where is is even slightly concentrated.

The manuscript is thus suitable for publication.

Reviewer #2: In this revised manuscript, the authors have done a nice job in carrying out additional modeling and MD simulation of N-P interaction to address my concern. They are also very responsive to the other review. Again, the paper addresses important issue on the topic of general interest to PLoS Pathogens readership and is ready for publication.

**Part II – Major Issues: Key Experiments Required for Acceptance**

Reviewer #1: None

Reviewer #2: There is no more concern.

**Part III – Minor Issues: Editorial and Data Presentation Modifications**

Reviewer #1: I was wondering if supplementary figure 5 could be included as a real figure as it gives important clues concerning STAT1 localization in RABV infected cells.

I would also suggest the authors to provide a fluorescence profile of both STAT 1 and N along a straight line crossing the Negri bodies presented in supplementary figure 5D. This would demonstrate the slight increase of STAT1 concentration in the Negri Bodies (or at least some of them).

Reviewer #2: No concerns.

PLOS authors have the option to publish the peer review history of their article (what does this mean?). If published, this will include your full peer review and any attached files.

Reviewer #1: No

Reviewer #2: No

---

## [Editor Report · Acceptance letter]

2 Jul 2021

Dear Professor Gooley,

We are delighted to inform you that your manuscript, "Definition of the immune evasion-replication interface of rabies virus P protein," has been formally accepted for publication in PLOS Pathogens.

Best regards,

Kasturi Haldar

Editor-in-Chief

PLOS Pathogens

orcid.org/0000-0001-5065-158X

Michael Malim

Editor-in-Chief

PLOS Pathogens

orcid.org/0000-0002-7699-2064